# Adaptive Fuzzy Sliding Mode Control of Omnidirectional Mobile Robots with Prescribed Performance

**Jeng-Tze Huang *** and **Chun-Kai Chiu**

Institute of Digital Mechatronic Technology, Chinese Culture University, Taipei 11114, Taiwan; joe1240chiu@gmail.com
* Correspondence: hzz4@ulive.pccu.edu.tw; Tel.: +886-2-28610511 (ext. 33711)

**Abstract:** Adaptive fuzzy sliding-mode control design for omnidirectional mobile robots with prescribed performance is presented in this work. First, an error transformation which transforms the constrained variable into an unconstrained one is carried out. Next, a fuzzy logic system (FLS) for approximating the unknown dynamics is constructed. Based on such a model, a nominal adaptive linearizing controller incorporating a serial-parallel model (SPM)-based composite algorithm, which improves the tracking performance of the overall closed-loop system, is synthesized. To solve the so-called "loss of controllability" problem, a smooth-switching algorithm is embedded which hands over the control authority to an auxiliary sliding-mode controller until the danger is safely bypassed. The proposed design ensures the semi-globally uniformly ultimately bounded stability of the closed-loop signals. Simulation works demonstrating the validity of the proposed design are presented in the final.

**Keywords:** omnidirectional mobile robot; adaptive fuzzy; composite algorithm; sliding-mode; smooth switching





## 1. Introduction

Omnidirectional mobile robots (OMR) are platforms that are able to move in any direction without reorientation. Owing to their great potential in a wide variety of industrial and military applications, the corresponding tracking control issues have attracted a great deal of attention over the decades [1–4]. The conventional PID controller, though still popular in industrial applications, may however exhibit discounted tracking performance when nonlinear dynamics or uncertainties are non-negligible. Regarding this, various nonlinear control schemes, such as feedback linearization [5], adaptive control [6], robust adaptive control [7–10], model-predictive control [11,12], etc., have been proposed to achieve better tracking performances in recent years. In particular, for an OMR with structured and unstructured uncertainties, the smooth robust adaptive control proposed in [10] was able to improve the tracking performances and avoid the control singularity simultaneously. However, all the aforementioned works rely heavily on the system modeling, which may be time-consuming or become difficult in complex environments.

In contrast, fuzzy logic controllers (FLCs) are able to translate the knowledge of human experts into robust control strategies without the need of a mathematical model of the system. Therefore, the numbers of fuzzy logic-based control algorithms for OMRs have been continuously increasing in the past few decades [13–17]. The work in [18] proposes a fuzzy controller for navigation and a behavioral method for obstacle avoidance. The works in [5,9,19] incorporated fuzzy logic with the popular PID control method for the path planning or trajectory tracking of OMRs. FLSs are used to approximate an ideal control law with an adaptive Mamdani-type fuzzy controller for an omnidirectional spherical mobile robot in [20]. Liu and Hsiao proposed an adaptive fuzzy sliding-mode tracking controller for OMRs [21]. Ren et al. presented a fuzzy-based intelligent obstacle-avoidance strategy

for a wheeled mobile robot in [22]. Despite these achievements, issues of tracking control for uncertain OMRs fulfilling a pre-specified output performance are seldom addressed.

Mobile robots are usually subjected to output/state constraints in reality, and various control schemes addressing such issues have been presented in the literature. To solve the aforementioned constraint problem, the so-called prescribed performance control (PPC) or the barrier Lyapunov function (BLF) techniques are generally adopted [23,24]. An adaptive NN-based control approach for wheeled mobile robots with full-state constraints was proposed in [24,25]. Considering the desired trajectory as a virtual servo constraint, an Udwadia–Kalaba-based adaptive robust control method is proposed for the trajectory tracking of an OMR in the presence of uncertainties [2]. The control of mobile robots with bounded torques is reported in [26], and control with velocity constraints is reported in [27,28]. In [12], a virtual-vehicle concept and an MPC strategy were combined to handle robot motion constraints and the path-following problems. However, issues of unknown input-gain functions and composite update algorithms were not taken into account in the aforementioned works.

Regarding this, a composite robust adaptive fuzzy control design for an OMR with prescribed performance is synthesized in this paper. To fulfill the prescribed performance imposed on the configuration variables, an error transformation that transforms the constrained problem into an unconstrained one is carried out first. Next, to render the adaptive control applicable, an FLS is invoked to approximate the unknown dynamics. Based on such a model, a nominal adaptive linearizing controller incorporating an SPM-based composite algorithm, which exhibits better tracking performance than the conventional Lyapunov-based update algorithm, is built. To conquer the so-called "loss of controllability" problem, a smooth-switching algorithm that hands over the control authority to an auxiliary sliding-mode controller is embedded. The major contributions of this paper are summarized as follows.

1.  The cases with an unknown control gain function and unknown nonlinearity arising from unmodeled dynamics, exogenous disturbances such as low-velocity friction [29], etc., are considered together at once;
2.  The FLC is invoked to approximate the lumped unknown nonlinearity, which renders the adaptive control easy to formulate;
3.  The PPC technique is incorporated to ensure the fulfillment of a prescribed performance requirement imposed on the configuration variables;
4.  The composite update algorithm is incorporated to improve the tracking performance further.

The paper is organized as follows: in Section 2, we state the problem and present some preliminaries. The adaptive sliding-mode control design that guarantees the prescribed performance is provided in Section 3. To verify the proposed design, simulation studies of an OMR are given in Section 4. Finally, we conclude in Section 5.

## 2. Preliminaries

### 2.1. Fuzzy Logic Systems

An FLS is an artificial decision-making system that emulates a human's reasoning processes. It is composed of four principal components: a fuzzifier, fuzzy rule base, inference engine, and defuzzifier. For the two-input/single-output system considered later in this paper, the fuzzy rules in the rule base will have the following forms [30]:

$$R^j : \text{IF } x_1 \text{ is } A_1^j \text{ and } x_2 \text{ is } A_2^j,$$
$$\text{THEN } y \text{ is } B^j \tag{1}$$

where $j = 1, 2, \cdots, m$ are the indices of the fuzzy rules, $x = [x_1, x_2]^T$ and $y$ are the input and output vectors of the FLS, respectively; and $A^j$ and $B^j$ are linguistic terms characterized by their membership functions $\mu_{A^j}(x_i)$ and $\mu_{B^j}(y)$, respectively.

By using a singleton fuzzifier, product inference, center-average defuzzifier, and Gaussian membership function, the output of the FLS can be expressed as linear combinations of fuzzy basis functions as follows:

$$y = \frac{\sum_{j=1}^{m} \bar{y}^j \left( \Pi_{i=1}^{n} \mu_{A_i^j}(x_i) \right)}{\sum_{j=1}^{m} \left( \Pi_{i=1}^{n} \mu_{A_i^j}(x_i) \right)}, \tag{2}$$

where $\bar{y}^j$ is the point at which $\mu_{B^j}$ has its maximum.

Define $\theta = [\bar{y}^1, \bar{y}^2, \cdots, \bar{y}^m]^T$ as the adjustable parameter vector and $\phi(x) = [\phi^1(x), \phi^2(x), \cdots, \phi^m(x)]^T$ as the corresponding fuzzy basis vector, where

$$\phi^j(x) = \frac{\Pi_{i=1}^{n} \mu_{A_i^j}(x_i)}{\sum_{j=1}^{m} \left( \Pi_{i=1}^{n} \mu_{A_i^j}(x_i) \right)} \tag{3}$$

Then, the output $y$ in (2) can be rewritten in a compact form as

$$y = \theta^T \phi(x) \tag{4}$$

Such a fuzzy logic system can be applied to approximate any continuous function $f(x)$ over a compact subset of input space to any given accuracy [30].

### 2.2. System Dynamics

This section introduces the kinematics and dynamics of an omnidirectional mobile platform with three independent driving wheels equally space 120 degrees apart. A photo of a real system is given in Figure 1. To describe the planar motion of a mobile platform, as shown in Figure 2, two coordinate frames—a moving frame $\{o_m, x_m, y_m\}$ located at the geometrical center of the cart and a stationary world frame $\{o_0, x_w, y_w\}$—are generally required. In the sequel, the world coordinate is selected as the default frame. Clearly, the motion of the platform can be decided if $p(t) = [x_p(t)\, y_p(t)]^T$ and $\theta(t)$ are specified.

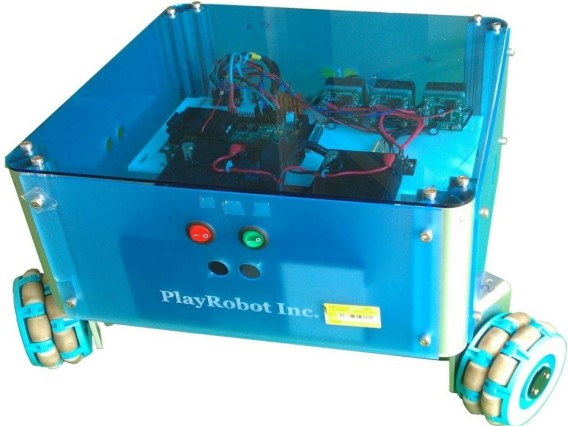

**Figure 1.** Three-dimensional plot of mobile robot.

Assuming non-slip conditions, the inverse kinematics are described by [31]

$$\omega = r^{-1} H(\theta) \dot{\xi} \tag{5}$$

where $\omega = [\omega_1, \omega_2, \omega_3]^T$ is the angular velocity of the wheels, $\xi = [x_p, y_p, \theta]^T$, and

$$H(\theta) = \begin{bmatrix} -S_\theta & C_\theta & l \\ -S_{\theta + \frac{2\pi}{3}} & C_{\theta + \frac{2\pi}{3}} & l \\ -S_{\theta - \frac{2\pi}{3}} & C_{\theta - \frac{2\pi}{3}} & l \end{bmatrix}, \tag{6}$$

with $S_\theta = \sin(\theta)$ and $C_\theta = \cos(\theta)$.

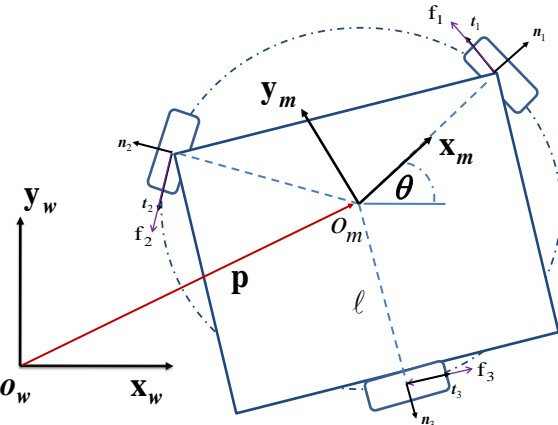

**Figure 2.** Coordinate system of an omnidirectional mobile robot.

It is noted that $H(\theta)$ is nonsingular for all $\theta \in R$ and hence is always invertible. By a direct calculation, we have

$$H^{-1} = \frac{2}{3} \begin{bmatrix} -S_\theta & -S_{\theta+\frac{2\pi}{3}} & -S_{\theta-\frac{2\pi}{3}} \\ C_\theta & C_{\theta+\frac{2\pi}{3}} & C_{\theta-\frac{2\pi}{3}} \\ \frac{1}{2l} & \frac{1}{2l} & \frac{1}{2l} \end{bmatrix} \tag{7}$$

The balances of linear and angular momentum result in [31]

$$m\ddot{\mathbf{p}}_c = \sum_{i=1}^{3} f_i \mathbf{t}_i,$$

$$J\ddot{\theta}\hat{z} = \sum_{i=1}^{3} lf_i(\mathbf{n}_i \times \mathbf{t}_i), \tag{8}$$

where $m$ and $J$ are the mass and the moment of inertia of the system, respectively, $\hat{z}$ is the unit vector of the $z$ axis, and $f_i$ are the three contact frictions of the wheels.

The torques $\tau_i$ and the driving voltages $u_i, i = 1, 2, 3$ for a DC motor are related as follows:

$$\tau_i = a_i u_i - b_i \omega_i, \quad i = 1, 2, 3 \tag{9}$$

where $a_i, b_i > 0, i = 1, 2, 3$ are the proportional constants.

Due to the non-slip condition, and by neglecting the wheel dynamics, we have

$$f_i = \frac{\tau_i}{r} \tag{10}$$

Define $u = [u_1 \, u_2 \, u_3]^T$. After a straightforward calculation, the dynamical equation can be written in a compact form:

$$M\ddot{\xi} + \frac{1}{r^2}H^T I_b H\dot{\xi} = \frac{1}{r}H^T I_a u + h(\xi, \dot{\xi}) \tag{11}$$

where $I_a = diag[I_i], I_b = diag[b_i], i = 1, 2, 3$, $h(\xi, \dot{\xi}) \in R^3$ is the the unknown nonlinearity arising from un-modeled dynamics, exogenous disturbances such as low-velocity friction [29], etc., and

$$M = \begin{bmatrix} m & 0 & 0 \\ 0 & m & 0 \\ 0 & 0 & J \end{bmatrix} \tag{12}$$

### 3. Control Design

Define the tracking error vector

$$e(t) = \xi(t) - \xi_d(t) \tag{13}$$

The paper aims to synthesize a control algorithm for u such that

(i)    The position vector $\xi(t)$ tracks the reference trajectory $\xi^d(t)$ accurately;
(ii)   The following prescribed tracking performance is achieved:

$$-\rho_i(t) < e_i(t) < \rho_i(t) \tag{14}$$

for all $t \geq 0$, where $1 \leq i \leq 3$ and $\rho_i(t)$ is a performance function given by

$$\rho_i(t) = (\rho_{0,i} - \rho_{\infty,i})e^{-\alpha_{0,i}t} + \rho_{\infty,i} \tag{15}$$

with $\rho_{0,i}, \rho_{\infty,i}, \alpha_{0,i}$ being positive constants at disposal.

To attain the aforementioned objectives, the following error transformation is carried out first [23]:

$$e_i(t) = \rho_i(t)T(\varepsilon_i), \quad i = 1, 2, 3 \tag{16}$$

where $\varepsilon_i$ is the transformed error and

$$T(\varepsilon_i) = \tanh(\varepsilon_i) \tag{17}$$

It is clear that item (ii) is equivalent to keeping $\varepsilon_i$ bounded for all time. On the other hand, the inverse transformation

$$\varepsilon_i = \tanh^{-1}\left(\frac{e_i}{\rho_i}\right) \tag{18}$$

is well-defined provided that (14) holds.

A direct differentiation of (18) yields

$$
\begin{aligned}
\dot{\varepsilon}_i &= \frac{1}{g_i}\left(\frac{\dot{e}_i}{\rho_i} - \frac{e_i}{\rho_i^2}\dot{\rho}_i\right) \\
\ddot{\varepsilon}_i &= \frac{2(e_i/\rho_i)}{g_i^2}\left(\frac{\dot{e}_i}{\rho_i} - \frac{e_i}{\rho_i^2}\dot{\rho}_i\right)^2 + \frac{1}{g_i}\left[\left(\frac{\ddot{e}_i}{\rho_i} - 2\frac{\dot{e}_i}{\rho_i^2}\dot{\rho}_i\right) + \left(\frac{2e_i}{\rho_i^3}\dot{\rho}_i^2 - \frac{e_i}{\rho_i^2}\ddot{\rho}_i\right)\right]
\end{aligned}
\tag{19}
$$

where $g_i = 1 - (e_i/\rho_i)^2$. We can start by defining the following error metric:

$$s = \dot{\varepsilon} + \lambda_s \varepsilon \tag{20}$$

where $\lambda_s > 0$ is the gain constant at disposal. By a direct differentiation of $s(t)$ in (20), and taking (19) into account, one has

$$
\begin{aligned}
\dot{s}_i = \frac{1}{g_i}\Big\{ &\frac{2(e_i/\rho_i)}{g_i}\left(\dot{e}_i - \frac{e_i}{\rho_i}\dot{\rho}_i\right)^2 + \left[\left(\ddot{e}_i - 2\frac{\dot{e}_i}{\rho_i}\dot{\rho}_i\right) + \left(\frac{2e_i}{\rho_i^2}\dot{\rho}_i^2 - \frac{e_i}{\rho_i}\ddot{\rho}_i\right)\right] \\
&+ \lambda_s\left(\frac{\dot{e}_i}{\rho_i} - \frac{e_i}{\rho_i^2}\dot{\rho}_i\right)\Big\}.
\end{aligned}
\tag{21}
$$

By virtue of the exponential stability of the $\varepsilon_1$ dynamics on the sub-manifold $s = 0$, the control of the original three dimensional system (11) is reduced to the stabilization problem of the one-dimensional $s$-dynamics in (21).

The $s$-dynamics in (21) can be written in a compact form:

$$\dot{s} = g^{-1}(Y_a^T \vartheta_a u + \eta + \psi), \tag{22}$$

where $g = diag[g_i]$, $Y_a^T \vartheta_a = r^{-1} M^{-1} H^T I_a$,

$$Y_a^T = \begin{bmatrix} -S_\theta & -S_{\theta + \frac{2\pi}{3}} & -S_{\theta - \frac{2\pi}{3}} & 0 & 0 & 0 \\ C_\theta & C_{\theta + \frac{2\pi}{3}} & C_{\theta - \frac{2\pi}{3}} & 0 & 0 & 0 \\ 0 & 0 & 0 & l & l & l \end{bmatrix},$$

$$\vartheta_a^T = \frac{1}{r} \begin{bmatrix} a_1 m & 0 & 0 & a_1 J & 0 & 0 \\ 0 & a_2 m & 0 & 0 & a_2 J & 0 \\ 0 & 0 & a_3 m & 0 & 0 & a_3 J \end{bmatrix}$$

$$\psi_i = \frac{2(e_i/\rho_i)}{g_i} \{ (\dot{e}_i - \frac{e_i}{\rho_i} \dot{\rho}_i)^2 + [-(\ddot{\xi}_{d,i} + 2\frac{\dot{e}_i}{\rho_i} \dot{\rho}_i) + (\frac{2e_i}{\rho_i^2} \dot{\rho}_i^2 - \frac{e_i}{\rho_i} \ddot{\rho}_i)]$$

$$+ \lambda_s (\dot{e}_i - \frac{e_i}{\rho_i} \dot{\rho}_i) \}, \quad i = 1, 2, 3, \tag{23}$$

and

$$\eta = M^{-1}(h - \frac{1}{r^2} H^T I_b H \dot{\xi}) \tag{24}$$

In this work, the FLSs are introduced to approximate the unknown function $\eta$ in (24). By virtue of the universal approximation property, given an a priori constant $\epsilon$,

$$\eta = \vartheta_b^T Y_b + \epsilon \tag{25}$$

where $\vartheta_b \in R^{n_b \times 3}$ and $Y_b \in R^{n_b}$ are the optimal weight vector and the regression matrix, respectively. The major advantage of such an approach is that the tedious procedure of determining the regression matrices indispensable to a standard adaptive controller is no longer required.

Substituting (25) into (22) yields

$$\dot{s} = g^{-1}(Y_a^T \vartheta_a u + \vartheta_b^T Y_b + \epsilon + \psi). \tag{26}$$

To stabilize (26), the following adaptive linearizing controller is frequently applied:

$$u_a = -[Y_a^T \hat{\vartheta}_a]^{-1}(k_s g s + \hat{\vartheta}_b^T Y_b + \psi). \tag{27}$$

As can be easily seen, such a control algorithm may suffer from control singularity when $det(Y_a \hat{\vartheta}_a) \to 0$. To prevent this occurrence, a smooth-switching algorithm in [10] is adopted. First, it is noted that $Y_a^T \vartheta_a H + H^T \vartheta^T Y_a = (M^{-1} H^T I_a H + H^T I_a H M^{-1})$ is positive-definite since

$$x^T (M^{-1} H^T I_a H + H^T I_a H M^{-1}) x \geq 2\lambda_{\min}(M^{-1}) \lambda_{\min}(H^T I_a H) |x|^2 > 0, \tag{28}$$

for all $x \in R^3, x \neq 0$.

There, then exists a $g_o > 0$ such that

$$\frac{1}{2}(g^{-1}x)^T (Y_a^T \vartheta_a H + H^T \vartheta^T Y_a)(g^{-1}x) \geq g_o |x|^2, \tag{29}$$

for all $x \in R^3, x \neq 0$.

The proposed control algorithm can now be specified as follows:

$$u = \varrho u_a + (1 - \varrho) u_s, \tag{30}$$

where

$$u_a = -[Y_a^T \hat{\vartheta}_a]^{-1}(k_s g s + \hat{\vartheta}_b^T Y_b + \psi),$$

$$u_s = -\frac{1}{g_o}(k_s + \frac{|g^{-1}(\hat{\vartheta}_b^T Y_b + \psi)|}{|s|}) H g^{-1} s, \tag{31}$$

with $k_s$ being the gain constant and $\varrho$ a smooth-switching function given by [10]

$$\varrho = 1 - \exp[-(\frac{\det[Y_a^T \hat{\vartheta}_a]}{w})^2]. \tag{32}$$

Basically, $w$ defines a region without a sharp boundary of the value $\det[Y_a^T \hat{\vartheta}_a]$, within which the sliding-mode control is in charge due to the approach of singularity. The control authority will be handed over to the nominal adaptive fuzzy controller until the singularity goes away. In addition to smoothness, the function $\varrho$ has two more desired properties:

(P1) $\lim_{\eta \to 0} \frac{\varrho(\eta)}{\eta} = 0$;

(P2) $v_s \overset{\Delta}{=} \sup_{\eta \neq 0} \left| \frac{\varrho(\eta)}{\eta} \right|$ is well defined.

By virtue of (P1), the singularity phenomenon will be totally avoided, while (P2) ensures the boundedness of the virtual control inputs for all time.

The following property is useful in the upcoming derivation and is quoted here for ease of reference [32]

$$|s(t)| \leq \epsilon \implies |\varepsilon(t)| \leq \frac{\epsilon}{\lambda_s}, \quad |\dot{\varepsilon}(t)| \leq 2\epsilon \tag{33}$$

Substituting (30) and (31) into (26) yields

$$
\begin{aligned}
\dot{s} &= g^{-1}\{Y_a^T \vartheta_a [\varrho u_a + (1-\varrho)u_s] + \vartheta_b^T Y_b + \epsilon + \psi\} \\
&= \varrho g^{-1}\{-(Y_a^T \vartheta_a)(Y_a^T \hat{\vartheta}_a)^{-1}(k_s g s + \hat{\vartheta}_b^T Y_b + \psi) + \vartheta_b^T Y_b + \psi\} \\
&\quad + (1-\varrho)g^{-1}\{-Y_a^T \vartheta_a \frac{1}{g_o}(k_s + \frac{|g^{-1}(\hat{\vartheta}_b^T Y_b + \psi)|}{|s|})Hg^{-1}s + \vartheta_b^T Y_b + \psi\} \\
&\quad + g^{-1}\epsilon \\
&= -\varrho(Y_a^T \tilde{\vartheta}_a)(Y_a^T \hat{\vartheta}_a)^{-1}(k_s g s + \hat{\vartheta}_b^T Y_b + \psi) + (1-\varrho)g^{-1} \\
&\quad \cdot \{-Y_a^T \vartheta_a \frac{1}{g_o}(k_s + \frac{|g^{-1}(\hat{\vartheta}_b^T Y_b + \psi)|}{|s|})Hg^{-1}s + \hat{\vartheta}_b^T Y_b + \psi\} \\
&\quad - k_s \varrho s + g^{-1}\tilde{\vartheta}_b^T Y_b + g^{-1}\epsilon
\end{aligned} \tag{34}
$$

When updating the estimated parameter vectors $\hat{\vartheta}_a$ and $\hat{\vartheta}_b$, it is known that the composite update algorithm generally results in better tracking performance [32]. Regarding this, the following serial–parallel estimation model is adopted in this paper:

$$\dot{\hat{s}} = g^{-1}(Y_a^T \hat{\vartheta}_a u + \hat{\vartheta}_b^T Y_b + \psi) + \beta\tilde{s}. \tag{35}$$

where $\hat{s}$ is the state of the SP estimation model, $\beta > 0$ is a gain constant, and $\tilde{s} = s - \hat{s}$ is the prediction error $\tilde{s}$.

By subtracting (26) from (35), the prediction error dynamics become

$$\dot{\tilde{s}} = g^{-1}(Y_a^T \tilde{\vartheta}_a u + \tilde{\vartheta}_b^T Y_b) - \beta\tilde{s} + g^{-1}\epsilon \tag{36}$$

The composite update algorithms for $\hat{\vartheta}_a$ and $\hat{\vartheta}_b$ can now be assigned as

$$
\begin{aligned}
\dot{\hat{\vartheta}}_a &= -\gamma_1[\varrho Y_a s(k_s g s + \hat{\vartheta}_b^T Y_b + \psi)^T (Y_a^T \hat{\vartheta}_a)^{-T} - \gamma_2 Y_a g^{-1}\tilde{s}u^T + \sigma_1 \hat{\vartheta}_a] \\
\dot{\hat{\vartheta}}_b &= \gamma_1[(Y_b s^T + \gamma_2 Y_b \tilde{s}^T)g^{-1} - \sigma_1 \hat{\vartheta}_b]
\end{aligned} \tag{37}
$$

Now, we are ready to state our main achievements as follows.

**Theorem 1.** *Consider the closed-loop system consisting of the system dynamics in (11), the control in (30), and the update algorithm in (37). If the value $g_o$ is known a priori and the gain constants are selected to satisfy $\beta > 1/2$, then the following points hold:*

- (T1): *All the signals in the closed-loop system remain bounded for all time;*
- (T2): *The tracking error $e_i(t)$ converges to the following set $\Omega_i$, which can be made arbitrarily small by increasing $\lambda_s$ and $k_v$ appropriately.*

$$\Omega_i = \{e_i \mid |e_i| \leq \rho_{\infty,i} \tanh(\sqrt{\frac{2d_v}{k_v \lambda_s^2}})\} \tag{38}$$

*where*

$$
\begin{aligned}
k_v &= \min(2k_s, \frac{2\beta - 1}{\gamma_2}, 2\gamma_1\sigma_1) \\
d_v &= \frac{1}{2}(|g^{-1}\epsilon|^2 + \sigma_1 tr[\vartheta_a \vartheta_a^T] + \sigma_1 tr[\vartheta_b^T \vartheta_b])
\end{aligned} \tag{39}
$$

**Proof of Theorem 1.** Select the following Lyapunov function

$$V(t) = \frac{1}{2}\{s^T s + \gamma_1^{-1}(tr[\tilde{\vartheta}_a \tilde{\vartheta}_a^T] + tr[\tilde{\vartheta}_b^T \tilde{\vartheta}_b]) + \gamma_2 \tilde{s}^T \tilde{s}\} \tag{40}$$

where $(\tilde{\cdot}) = (\cdot) - (\hat{\cdot})$.

The time derivative of $V(t)$ along the system dynamics in (11) can be calculated as follows:

$$
\begin{aligned}
\dot{V}(t) = {}& s^T \dot{s} + \gamma_2 \tilde{s}^T \dot{\tilde{s}} + \gamma_1^{-1}(tr[\tilde{\vartheta}_a \dot{\tilde{\vartheta}}_a^T] + \tilde{\vartheta}_b^T \dot{\tilde{\vartheta}}_b) \\
= {}& s^T\{-\varrho(Y_a^T \tilde{\vartheta}_a)(Y_a^T \hat{\vartheta}_a)^{-1}(k_s g s + \hat{\vartheta}_b^T Y_b + \psi) - k_s \varrho s + g^{-1}\tilde{\vartheta}_b^T Y_b\} \\
& + (1 - \varrho)\{\frac{-1}{2g_o}s^T g^{-1}(Y_a^T \vartheta_a H + H^T \vartheta_a^T Y_a)g^{-1}s(k_s + \frac{|g^{-1}(\hat{\vartheta}_b^T Y_b + \psi)|}{|s|}) \\
& + s^T g^{-1}(\hat{\vartheta}_b^T Y_b + \psi)\} + s^T g^{-1}\epsilon + tr[\tilde{\vartheta}_a\{\varrho(Y_a^T \hat{\vartheta}_a)^{-1}(k_s g s + \hat{\vartheta}_b^T Y_b + \psi)s^T Y_a^T \\
& - \gamma_2 u \tilde{s}^T g^{-1} Y_a^T + \sigma_1 \hat{\vartheta}_a^T\}] - tr[\tilde{\vartheta}_b^T((Y_b s^T + \gamma_2 Y_b \tilde{s}^T)g^{-1} - \sigma_1 \hat{\vartheta}_b)] \\
& + \gamma_2 \tilde{s}^T[g^{-1}(Y_a^T \tilde{\vartheta}_a u + \hat{\vartheta}_b^T Y_b) - \beta \tilde{s} + g^{-1}\epsilon]
\end{aligned} \tag{41}
$$

Note that

$$
\begin{aligned}
& -\varrho s^T(Y_a^T \tilde{\vartheta}_a)(Y_a^T \hat{\vartheta}_a)^{-1}(k_s g s + \hat{\vartheta}_b^T Y_b + \psi) + tr[\tilde{\vartheta}_a\{\varrho(Y_a^T \hat{\vartheta}_a)^{-1} \\
& \cdot (k_s g s + \hat{\vartheta}_b^T Y_b + \psi)s^T Y_a^T\}] = 0, \\
& s^T g^{-1}\tilde{\vartheta}_b^T Y_b - tr[\tilde{\vartheta}_b^T Y_b s^T g^{-1}] = 0, \\
& -\gamma_2 tr[\tilde{\vartheta}_a u \tilde{s}^T g^{-1} Y_a^T] + \gamma_2 \tilde{s}^T g^{-1} Y_a^T \tilde{\vartheta}_a u = 0, \\
& -\gamma_2 tr[\tilde{\vartheta}_b^T Y_b \tilde{s}^T g^{-1}] + \gamma_2 \tilde{s}^T g^{-1}\tilde{\vartheta}_b^T Y_b = 0, \\
& -\frac{1}{2g_o}(k_s + \frac{|g^{-1}(\hat{\vartheta}_b^T Y_b + \psi)|}{|s|})s^T g^{-1}(Y_a^T \vartheta_a H + H^T \vartheta_a^T Y_a)g^{-1}s \\
& \leq -k_s s^T s - |g^{-1}(\hat{\vartheta}_b^T Y_b + \psi_b)||s|
\end{aligned} \tag{42}
$$

Substituting (42) into (41) yields

$$
\begin{aligned}
\dot{V}(t) \leq {}& -k_s s^T s + \sigma_1 tr[\tilde{\vartheta}_a \hat{\vartheta}_a^T] + \sigma_1 tr[\tilde{\vartheta}_b^T \hat{\vartheta}_b] - \gamma_2 \beta \tilde{s}^T \tilde{s} \\
& + s^T g^{-1}\epsilon + \gamma_2 \tilde{s}^T g^{-1}\epsilon
\end{aligned} \tag{43}
$$

Next, by completing the squares,

$$
\begin{aligned}
\tilde{s}^T g^{-1}\epsilon &\leq \frac{1}{2}(-\tilde{s}^T \tilde{s} + |g^{-1}\epsilon|^2) \\
\sigma_1 tr[\tilde{\vartheta}_a \hat{\vartheta}_a^T] &\leq \frac{\sigma_1}{2}(-tr[\tilde{\vartheta}_a \tilde{\vartheta}_a^T] + tr[\vartheta_a \vartheta_a^T]) \\
\sigma_1 tr[\tilde{\vartheta}_b^T \hat{\vartheta}_b] &\leq \frac{\sigma_1}{2}(-tr[\tilde{\vartheta}_b^T \tilde{\vartheta}_b] + tr[\vartheta_b^T \vartheta_b])
\end{aligned} \tag{44}
$$

Substituting (44) into (43) yields

$$\dot{V}(t) \leq -k_v V(t) + d_v, \tag{45}$$

Apparently V(t) is bounded for $\dot{V}(t) \leq 0$ when $V(t) \geq d_v/k_v$; the boundedness of the signals $s(t)$, $\tilde{s}(t)$, $\hat{\vartheta}_a(t)$, and $\hat{\vartheta}_b(t)$ follows directly by definition. This in turn ensures the boundedness of the error signals $e(t)$ and $\dot{e}(t)$ in (13) and the controllers in (30). Therefore, T1 is proven.

On the other hand, (45) also implies that

$$\frac{1}{2}s^T s \quad \leq \quad V(0)e^{-k_v t} + \frac{d_v}{k_v}(1 - e^{-k_v t}) \tag{46}$$

It is easy to see that $|s(t)|$ eventually converges to a small set with radius $\sqrt{2d_v/k_v}$. Based on (33), $|\varepsilon(t)|$ eventually converges to a set with radius $\sqrt{2d_v/k_v}/\lambda_s$, which in turn implies that $e(t)$ converges to $\Omega_e$ as $t \to \infty$. $\quad \square$

**Remark 1.** *As is well known, the sign function appearing in (31) can be replaced with the saturation function to eliminate the possible chattering behavior [32].*

## 4. Simulation

Based on the pseudo code listed in Algorithm 1, simulation results are presented in this section to demonstrate the validity of the proposed design.

The desired trajectory is a circle given by $\xi_d = 0.5[\cos(0.4t), \sin(0.4t), 0]^T$. The unknown nonlinearity is as follows: $h(\xi, \dot{\xi}) = [\xi_2 \xi_3 \cos(\xi_1 + \dot{\xi}_1 + \dot{\xi}_2 \dot{\xi}_3), \sin(\xi_1 + 2)\xi_2 + \xi_3 \cos(\xi_2 + \dot{\xi}_1) + \dot{\xi}_2 \dot{\xi}_3, \dot{\xi}_1 \dot{\xi}_2 \dot{\xi}_3/(2 + \sin(\xi_1 \xi_2 + \xi_3 + 2))]^T$. The approximations of $\eta_i(\xi, \dot{\xi})$, $i = 1, 2, 3$ are conducted within the following fuzzy region:

$$\Omega \quad = \quad \{(\xi, \dot{\xi})| - 0.6 \leq \xi_p \leq 0.6, -0.5 \leq \dot{\xi}_p \leq 0.5, p = 1, 2, 3\} \tag{47}$$

The membership functions for the input variables $\xi_p$ and $\dot{\xi}_p$ are given respectively by

$$\mu_{\xi_p}^k \quad = \quad \exp[-(\frac{\xi_i - 0.3(k-3)}{0.45})^2], \quad 1 \leq k \leq 5$$

$$\mu_{\dot{\xi}_p}^l \quad = \quad \exp[-(\frac{\dot{\xi}_j - 0.5(l-2)}{0.75})^2], \quad 1 \leq l \leq 3 \tag{48}$$

The corresponding fuzzy basis function $\phi^j$ in (3) can then be easily constructed. The remaining numerical values adopted in this simulation are $m = 10.0$ kg, $J = 0.5$ kg m$^2$, $r = 0.05$ m, $l = 0.4$ m, $a_1 = 0.2, a_2 = 0.3, a_3 = 0.4, b_1 = 0.01, b_2 = 0.015, b_3 = 0.02$, $\lambda_s = 1.0, g_o = 0.1, \delta = 0.5, \rho_0 = [0.5, 0.5, 0.5]^T, \rho_\infty = [0.05, 0.05, 0.05]^T, \alpha_0 = [0.5, 0.5, 0.5]^T$, $k_s = 1.0, \gamma_1 = \gamma_2 = 1.0$, and $\sigma_1 = 0.1$. The initials are randomly generated within $[0, 0.2]$.

The actual trajectory of the omnidirectional mobile robot follows the commanded circle faithfully, as can be seen from Figure 3. The corresponding tracking errors shown in Figure 4 converge quickly to a small set around zero while fulfilling the prescribed performance (black dashed lines) at the same time. The switching signal in Figure 5 indicates that the nominal adaptive fuzzy controller dominates during the first few seconds and then switches the control authority to the sliding-mode controller due to the quick decay of the estimated parameter vector $\hat{\vartheta}_a(t)$ to a small value. The control inputs depicted in Figure 6 are rather mild except for the initial peaks due to the large tracking errors therein.

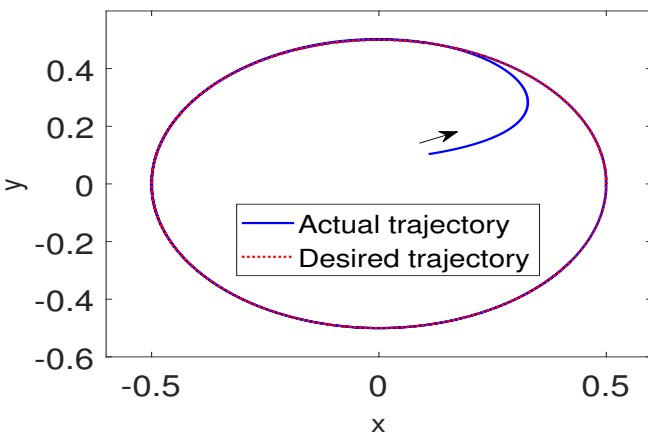

**Figure 3.** Actual trajectory of an mobile cart following a circle.

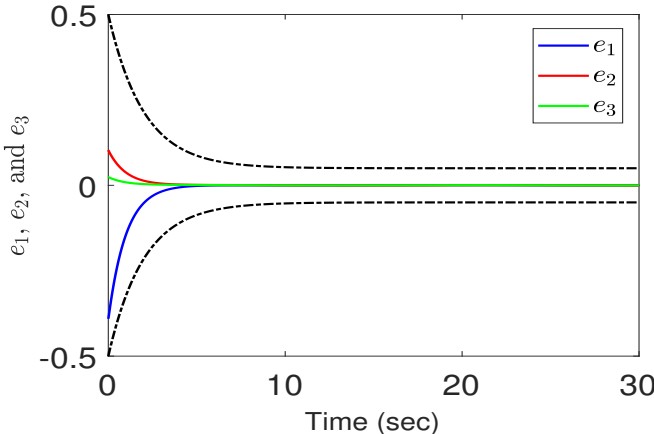

**Figure 4.** Tracking errors vs. time under a PPC in Case 1.

---

**Algorithm 1:** Pseudo code for the proposed control algorithm.

---

**Start**
**Initialization**: PPC parameters $\rho_0, \rho_\infty, \alpha_0$; FLS parameters in $\mu_{A^j}, \mu_{B^j}, j = 1, \cdots, m$;
control and tuning gains $\lambda_s, k_s, \gamma_1, \gamma_2, \sigma_1$
1 **For** $i = 1 : n$
2 % Read the sensor inputs $\omega_i$, $i = 1, 2, 3$
3 % Calculate the forward kinematics
4 $\dot{\xi} = rH^{-1}(\theta)\omega$,
5 $\xi = \int \dot{\xi} dt$
6 % Calculate the transformed error $\varepsilon_i$ in (18) and $\dot{\varepsilon}_i$ in (19)
7 % Calculate the error metric $s$ in (20)
8 % Calculate the control input $u(t)$ in (30)
9 % Calculate the composite update algorithms $\dot{\hat{\vartheta}}_a$ and $\dot{\hat{\vartheta}}_b$ in (35)
10 **End For** $i$

---

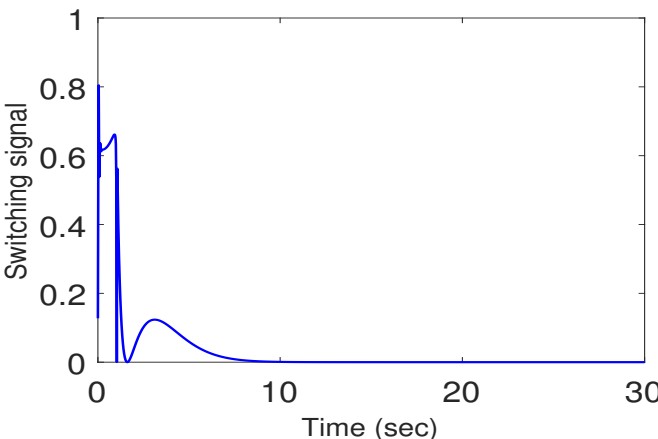

**Figure 5.** Switching signal in Case 1.

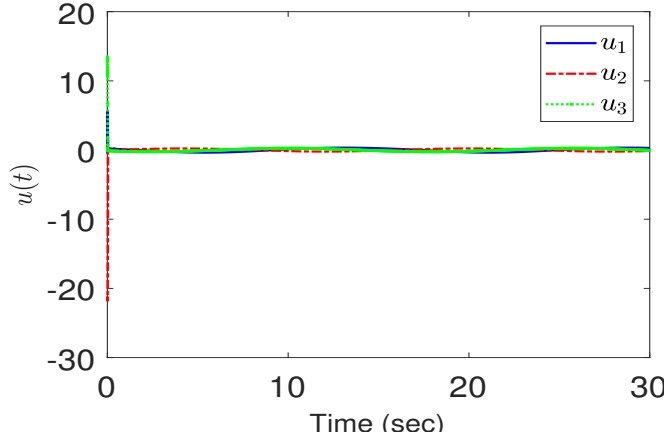

**Figure 6.** Control effort in Case 1.

Next, to see how different prescribed performances may affect the tracking performance and the control efforts, we change the corresponding parameters as follows: $\rho_\infty = [0.02, 0.02, 0.02]^T$, $\alpha_0 = [1.0, 1.0, 1.0]^T$, The tracking performance is depicted in Figure 7. As can be expected, the perfromance is better than those in Figure 4 with the price of a larger control effort in the beginning, as shown in Figure 8.

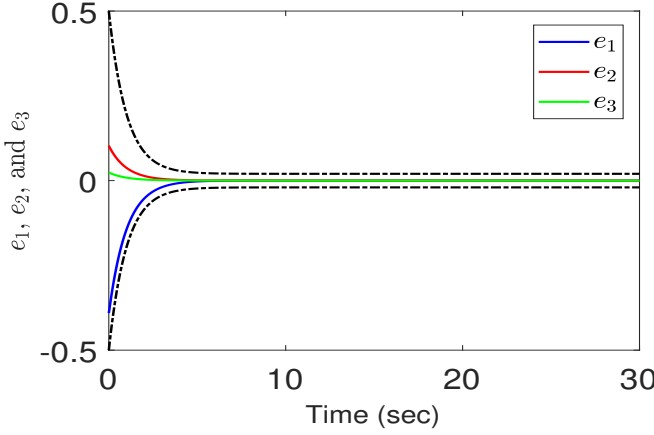

**Figure 7.** Tracking errors vs. time under a PPC in Case 2.

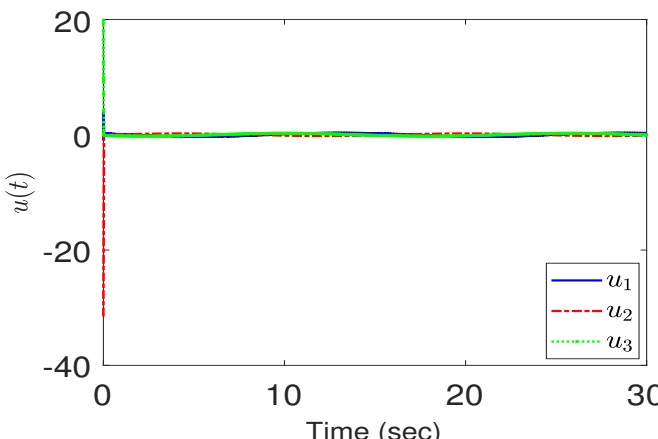

**Figure 8.** Control effort in Case 2.

## 5. Conclusions

We have constructed a fuzzy sliding-mode controller to achieve the objectives of the trajectory tracking of a three-wheeled uncertain omnidirectional mobile robot system under a prescribed performance constraint. The proposed design ensures asymptotical tracking stability and avoids the control singularity at the same time. Compared with conventional control methodologies, such an approach exhibits the major advantage of avoiding the tedious procedure of determining the regression matrices. Extensions to more difficult tasks such as slipping avoidance are currently under study.

**Author Contributions:** J.-T.H.: methodology, C.-K.C.: software. All authors have read and agreed to the published version of the manuscript.

**Funding:** This research was funded by Ministry of Science and Technology, Taiwan, grant number MOST 110-2221-E-034-014.

**Conflicts of Interest:** The authors declare no conflict of interest.

## Abbreviations

The following abbreviations are used in this manuscript:

| | |
|---|---|
| FLS | Fuzzy logic system |
| SP | Serial–parallel |
| OMR | Omnidirectional mobile robot |
| PPC | Prescribed performance control |

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
