# Peer review of "Adaptive Fuzzy Sliding Mode Control of Omnidirectional Mobile Robots with Prescribed Performance"

_processes, doi:10.3390/pr9122211_

Round 1
Reviewer 1 Report
An interesting approach to control a of omnidirectional mobile robots with prescribed performance, which requires a more detailed description.
- You propose to use the FLS presented by Wang L. X. [29] in your references and described by equation (2). It would be useful for the reader if you could explain what [y1, y2, ...ym] represent.
- Explain why it is important to introduce ζi in equation (17).
- Explain why it is important to introduce the new error metric s in equation (19).
- Is the sentence "By substituting (24) into (20), it yields" in line 124 correct? It should probably be " ... into 20".
- Equation (36) is the core of the adaptive fuzzy algorithm. Explain where FLS (2) is used to approximate the unmodelled dynamics. Where are the parameters of the FLS to be adapted?
- Draw a flowchart of the entire algorithm used in the simulation study or write pseudo code for all the necessary calculations. This would help the reader to understand the functioning of the proposed design of the fuzzy sliding-mode controller.
Reviewer 2 Report
In this paper, a fuzzy logic system is applied to approximate the unknown dynamics of the mobile robot. As the control method, a normal sliding mode controller with the conventional prescrbing function method was design to enhence the tracking performance of the mobile robot system.
Comments for this pape are listed as follows:
1. The reason for applying FLS is not clear. The given dynamics of the mobile robot in (23) is not complex.
2. The sliding surface in (18) and controller in (29) are common and new method are not found.
3. The prescrbing function method given in (12) - (17) is well known one and more contributions are not presented.
Thus, the presented control system is combination of the conventional methods and much efforts for new contribution are required.
Round 2
Reviewer 2 Report
In the revised submission, the remarkable contributions were not found yet.
The proposed method seems to the mixed version by utilizing traditional FLS, SMC, prescribing function, and adative laws. At least, a new method is required to enhence the quality of the paper. In this submission, regretfully, this is not found. More improved method for FLS, SMC, and precribing function is needed to be published in this journal.